# Higher fibrinogen-to-albumin ratio is associated with the severity of toxin associated acute kidney injury in high-altitude (3650 m) population: A retrospective analysis

Wenling Yang[1,2☯*], Lixia Cui[2☯], Yan He[2], Yong A[2], Jianbin Zhang[2], Quzhen Suolang[2], Luobu Ciren[2], Lei Zhang[2]

1 Department of Nephrology, Peking University Third Hospital, Beijing, People's Republic of China,
2 Department of Nephrology, Tibet Autonomous Region People's Hospital, La-Sa, The Tibet Autonomous Region, People's Republic of China

☯ These authors contributed equally to this work.
* 0163171518@bjmu.edu.cn

## Abstract

### Background

Few reports used acute kidney injury concept in the area of toxic kidney damage, especially in highland areas. Recent evidence suggests that the fibrinogen-to-albumin ratio is significantly associated with the incidence and severity of acute kidney injury, highlighting its potential clinical utility for risk stratification and prognostication in high-risk populations. We investigated the clinical characteristics and risk factors of toxic kidney injury in highland areas using the acute kidney injury criteria defined by the kidney disease improving global outcomes work group, with focus on the significance of the fibrinogen- to- albumin ratio.

### Methods

Clinical data of poisoned patients who had electronic inpatient medical records during the past ten years were retrospectively analyzed, and risk factors affecting renal function were investigated.

### Results

Sixty-five inpatients fulfill the criteria with the median age of 36.7 years old, including 40 females (61.5%). Thirty-four patients (52.3%) underwent blood purification, mainly hemoperfusion (n = 33). Medicine poisoning accounts for 53.8% (n = 35), including antipsychotic sedatives (n = 14) and Tibetan medicine (n = 9). Pesticide poisoning ranks the second (27.7%, n = 18). And the last was toxic substances eaten as food (n = 12). Among them, 39 cases (60.0%) were attributed to intentional self-poisoning. Multivariate linear regression analysis revealed that the serum creatinine concentration was positively related to the fibrinogen-to-albumin ratio, and to the

which permits unrestricted use, distribution, and reproduction in any medium, provided the original author and source are credited.

**Data availability statement:** The minimal data is uploaded as Supporting Information files.

**Funding:** This work was supported in part by a research Initiation Fund Sponsored by Peking University Third Hospital for Excellent Returned Overseas Scholars (BYSYLXHG2019007). YWL received the award. The funders had no role in study design, data collection and analysis, decision to publish, or preparation of the manuscript.

**Competing interests:** The authors report there are no competing interests to declare.

concentrations of uric acid and serum phosphate: The reciprocal of the highest serum creatinine concentration (µmol/L) = 0.032–0.002 * uric acid concentration (µmol/L)/100 - 0.005 * serum phosphate concentration (mmol/L) −0.078 * the fibrinogen-to-albumin ratio + ε. The study demonstrated a dose-dependent association between elevated fibrinogen-to-albumin ratio and increased serum creatinine levels.

## Conclusions

The public needs to be well informed to minimize the chance of exposure to excessive medicine, wild vegetables, toxic mushrooms, and pesticides. Traditional Tibetan medicine, unique to this plateau region, requires professional guidance for its identification, processing, and dosage. Mental well-being in plateau areas must be prioritized, and regulatory oversight of pesticides and antipsychotic sedatives need to be strengthened to mitigate the risks of overdose and toxic exposure. The serum creatinine concentration in poisoned patients in plateau regions may be affected by the fibrinogen-to-albumin ratio. Therapy against microinflammation, higher uric acid and phosphate levels may prevent further kidney injury.

## Introduction

The causes and severity of poisoning differ widely among regions [1,2]. Fish gall cysts and mushroom poisoning are more common in the southern region, and organophosphorus pesticide poisoning is more common in the northern region. Some patients have only digestive symptoms, and some have severe liver and kidney damage or even die [3–5]. Acute kidney injury (AKI) is a prognostic factor in poisoned patients [6]. However, the definition of AKI varies widely across the literature prior to 2012. Previously described renal damage includes proteinuria, hematuria and renal dysfunction. Some even found no significant difference in blood urea concentrations between the non-surviving and surviving groups [7]. Since AKI was updated in 2012 [8] by the Kidney Disease: Improving Global Outcomes (KDIGO) work group, few reports have used the AKI criteria by KDIGO in the area of toxic kidney injury, especially in highland areas [3,4,7].

In this study, we conducted a retrospective analysis of the clinical characteristics and risk factors of toxic kidney injury using the AKI criteria by KDIGO in inpatients from highland areas. Emerging evidence suggests that fibrinogen-to-albumin ratio (FAR), as a novel inflammatory biomarker, is significantly associated with AKI, potentially mediated through its reflection of microinflammatory states, oxidative stress pathways, and endothelial dysfunction implicated in both glomerular and tubular damage [9]. Chronic exposure to hypobaric hypoxia at high altitudes was reported to increase plasma fibrinogen level [10,11], which suggest it might elevate FAR. Hypoxia promotes fibroblast activation in renal tissue, together with inflammatory cell recruitment and tubular epithelial cell damage, leads to renal tubule interstitial fibrosis [12]. We hypothesized that FAR would predict AKI severity independent of traditional inflammatory markers in plateau area.

## Methods

Clinical data of poisoned inpatients fulfilling the criterion were retrospectively analyzed during July 20th 2022 to July 20th 2023 to determine the clinical characteristics and risk factors of renal damage in the plateau area. The study was approved by the Human Research Ethics Committee in Tibet Autonomous Region People's Hospital (ME- TBHP- 2022- KJ- 038). This is a retrospective study and the data were analyzed anonymously, so informed consent is spared by the Ethics Committee in Tibet Autonomous Region People's Hospital.

Inclusion criterion: All poisoned patients who had been hospitalized consecutively to our wards and had electronic medical records available in the past ten years were enrolled.

Exclusion criterion: The patients who had neither urine volume record nor renal function data during the illness were excluded.

AKI status was defined as literature [8]. An increase in serum creatinine (SCr) concentration from baseline can be substituted as a decrease from the baseline during follow-up if there are no prior tests. A urine output less than 400 mL/day may be an alternative to ≤ 0.5 mL/ (kg/h) for 6 hours. For those with only one test of serum creatinine, we made diagnosis of AKI with reference to the criteria of urine volume and clinical diagnosis.

All analysis was performed with SPSS 26.0 (IBM Corp., USA). Categorical variables are compared with chi-square tests, corrected for continuity if it is a 2* 2 table or Fisher's exact test if >20% of cells have the expected count < 5 in a table larger than 2×2. Continuous variables are compared between two groups with independent sample t tests or Mann–Whitney U tests according to the data distribution. To make full use of the data with missing values such as total bilirubin and hemoglobin etc, we used the averaged value of that at disease onset and that at discharge in the comparison analysis. Those measured at emergency department before any treatments were regarded as data at onset. Pairwise deletion was utilized in correlation assessments. In the multivariate linear regression analysis, we designated the highest SCr value rather than the value at onset or discharge as the dependent variable. Variables were sequentially entered into three hierarchical models: step 1, FAR as an inflammatory biomarker; step 2, FAR combined the demographic parameters; step 3, the above integrated laboratory-derived variables. All the variables with significant relationship with the highest SCr concentration in the spearman bivariate correlation analysis and variables with significant difference between AKI and non-AKI group, excluding those with possible collinearity, were investigated further in the multivariate linear regression analysis as independent variables.

All the statistical tests were 2-sided. A P value less than 0.05 was considered statistically significant.

## Results

Electronic medical records were available since September 2013. So, we can only obtain the medical records since 2013. A total of 65 inpatients fulfilled the criteria, while one patient was excluded due to lack of both renal function and urine volume record. The median age was 36.7 years old at admission. Forty patients were female (61.5%). All the participants resided at an altitude of 3650.0 (2200.0, 3760.9) meters above sea level.

Thirty-five patients (53.8%) were poisoned by medication. Among them, fourteen cases (21.5%) were attributed to antipsychotic sedatives, including benzodiazepines (n = 11, such as estazolam (n = 7) and alprazolam (n = 3)), antipsychotic drugs (n = 4, such as quetiapine, olanzapine and risperidone) and antianxiety/ antidepressive drugs (n = 4). The remaining twenty-one patients (32.3%) took other medications, including Tibetan medicine (traditional herbs) (n = 9), antiepileptic drugs (n = 4, such as phenobarbital, carbamazepine, lamotrigine), antibiotics (n = 3, such as metronidazole, cephalexin, rifampicin), nonsteroidal anti-inflammatory drugs (n = 1), and others (n = 4). In summary, nine patients (25.7%) took two or more kinds of medication while the majority used a combination with benzodiazepine sedative drugs, and two patients (5.7%) took medication after drinking.

The second type of poisoning comes from pesticides (n = 18, 27.7%), including insecticides (n = 7), herbicides (n = 4), rodenticides (n = 3) and other organophosphorus pesticides (n = 4).

The third type of poisoning were toxic substances eaten as food (n = 12, 18.5%), which involved the consumption of poisonous mushrooms (n = 4), wild vegetables (n = 5), smelly field snails/grilled fish (n = 2), and stramonium seeds (n = 1). Except for one person who committed suicide by taking stramonium seeds, everyone else ate the poisons by mistake.

Thirty-nine patients (60.0%) were attributed to intentional self-poisoning, 20 (30.8%) resulted from accidental ingestion, and in the remaining cases (9.2%), the reasons for poisoning could not be determined conclusively. Further analysis of the 39 patients with intentional self-poisoning demonstrated the main types of poisons were pesticides (38.5%, n = 15) and medication (59.0%, n = 23), while antipsychotic sedatives (60.9%, n = 14) were the main type of medicine. Meanwhile, we found 48.7% (n = 19) among the 39 patients had no pre-existing medical conditions and 38.5% (n = 15) had chronic multi-morbidity inclusive of mental health disorders. All the poisons were taken into the body orally.

Among the participants, 33 patients (50.8%) were unable to accurately estimate the dosage of poison. Six patients (9.2%) ingested a poison dose ≤ 10 capsules or 10 milliliters (mL), while 12 patients (18.5%) consumed a dose ranging from 11 to 50 capsules or mL. Additionally, there were 14 patients (21.5%) in whom the poison dose exceeded >50 capsules or mL. Notably, no significant association was detected between the categories of toxic substance dosage and mental disorder ($\chi^2$ = 2.595, P = 0.501) or AKI or not ($\chi^2$ = 0.385, P > 0.05).

Mental disorders occurred in thirty-one patients (47.7%), including coma (n = 17, 54.8%), confusion (n = 8), drowsiness (n = 4) and restlessness (n = 2; both due to laryngeal edema). Altogether, twenty-four serious patients (36.9%) presented with multiple organ failure, shock, coma or cardiac arrest. No significant association was detected between the categories of toxin dosage and disease severity ($\chi^2$ = 2.127, P = 0.573).

Thirty-one patients (47.7%) presented with one or more comorbidities including infections (n = 14), psychiatric disorders (n = 8), chronic anemia (n = 3), hypertension (n = 3), diabetes mellitus (n = 2), sequelae of cerebral infarction (n = 1), pelvic fracture (n = 1) and other miscellaneous conditions (n = 7).

The duration from poisoning to emergency room was 12.0 (4.0, 24.0) hours. A majority of patients (61.5%) were transferred to emergency in more than 6 hours. In addition to routine gastric lavage, catharsis, symptomatic support treatment and specific detoxification if necessary, thirty-four patients (52.3%) underwent blood purification. Hemoperfusion (n = 33) was carried out with an HA230 single-use hemoperfutor for 2 hours each treatment session with a blood flow of 200 ml/min. Sixteen patients (48.5%) had two times of hemoperfusion, and eleven patients (33.3%) had three or four treatments, administered once per day. All patients were discharged with improvement. The length of hospital stay was 4.0 (3.0, 9.0) days (1–31 days).

Six patients (9.2%) had no urinalysis. Urine protein and glucose were detected in 15 patients (25.4%) and 13 patients (22.0%) respectively. Urine output was measured in 55 patients. Nineteen patients (34.5%) had a urine output greater than 3500 mL/day, and five had a urine output ≤ 400 mL/day.

Twelve patients (18.5%) were defined as having AKI. Compared with those in the non-AKI group (Table 1), the patients in the AKI group had higher urea and creatinine concentrations, higher blood pressure, higher heart rate, greater NLR, longer length of hospital stay, lower lymphocyte percentage and less urine output daily. More patients in the AKI group had proteinuria. If we defined AKI without regard to urine output, then only 8 patients had AKI (Table 1 in S1 Tables). The white blood cell count, neutrophil percentage, NLR, fibrinogen, D-dimer, FDP, FAR, AST, and ALT were greater in the AKI group than in the non-AKI group, in addition to the greater serum urea and creatinine concentrations (all P < 0.05). Blood pressure and average heart rate were greater in AKI patients than in non-AKI patients. Additionally, we can see that AKI patients were more likely to have proteinuria and longer hospital stays.

Spearman bivariate correlation analysis (Table 2 in S1 Tables) demonstrated that the highest serum creatinine concentration was correlated with the urea concentration at onset, sex, hemoglobin, uric acid, serum myoglobin and phosphate (r values were 0.307–0.485 respectively) and may be correlated (p < 0.2) with the FAR, age, average total bilirubin (TBIL), serum calcium concentration and proteinuria or not. The scatter plot of FAR vs reciprocal of the highest SCr during the disease was shown in Fig 1.

**Table 1. Comparison of demographic and laboratory data between the two groups of poisoned patients with or without AKI.**

| Variables | n (%) | Mean±SD/ median (25%, 75% quartiles) | AKI (n=12) | Non-AKI (n=53) | Statistics (T value/ Mann–Whitney Z value/ $\chi^2$) | Sig. (two-tailed) |
|---|---|---|---|---|---|---|
| Age, years | 65 (100.0%) | 36.7 (29.1, 49.6) | 45.3 (28.7, 56.0) | 36.0 (29.1, 46.9) | 1.023 | 0.306 |
| Male, n (%) | 25 (38.5%) | / | 8 (66.7%) | 17 (32.1%) | 3.593 | 0.058 |
| Hemoperfusion, n (%) | 33 (50.8%) | / | 8 (66.7%) | 25 (47.2%) | 0.81 | 0.368 |
| Proteinuria, n (%) | 15 (25.4%) (n=59) | / | 6 (54.5%) (n=11) | 9 (18.8%) (n=48) | 4.307 | 0.038 |
| Duration between poisoning to the clinic, hours | 62 (95.4%) | 12.0 (4.0,24.0) | 8.0 (5.0, 24.0) (n=11) | 12.0 (4.0, 24.0) (n=51) | −0.065 | 0.948 |
| Length of stay, days | 65 (100.0%) | 4.0 (3.0, 9.0) | 6.5 (4.5, 15.0) | 4.0 (3.0, 7.0) | 2.397 | 0.017 |
| Average altitude, m | 65(100.0%) | 3650.0(2200.0, 3760.9) | 3575.0 (100.0, 4094.1) | 3650.0 (2919.0, 3694.0) | 0.681 | 0.496 |
| Lowest urine output, mL/d | 53 (81.5%) | 1020.0 (650.0, 1600.0) | 470.0 (275.0, 825.0) (n=11) | 1150.0 (750.0, 1670.0) (n=42) | −2.678 | 0.007 |
| Serum urea, mmol/L | 59 (90.8%) | 4.3 (3.5, 6.4) | 6.5 (4.1, 9.2) (n=11) | 4.2 (3.2, 5.9) (n=48) | 2.112 | 0.035 |
| Serum creatinine, µmol/L | 61 (93.8%) | 66.8 (52.0, 82.0) | 93.7 (70.3, 144.0) | 62.2 (49.0, 76.4) (n=49) | 3.103 | 0.002 |
| Serum uric acid, µmol/L | 44 (67.7%) | 302.9±117.8 | 353.1±99.2 (n=9) | 290.0±120.0 (n=35) | 1.453 | 0.154 |
| Serum potassium, mmol/L | 65 (100.0%) | 3.7 (3.3, 4.1) | 4.0 (3.6, 4.5) | 3.7 (3.3, 4.0) | 1.346 | 0.178 |
| $CO_2CP$, mmol/L | 59 (90.8%) | 16.8±3.0 | 16.7±3.1 | 16.8±3.0 (n=47) | −0.115 | 0.908 |
| Serum calcium, mmol/L | 54 (83.1%) | 2.2±0.2 | 2.2±0.3 (n=9) | 2.1±0.2 (n=45) | 0.905 | 0.389 |
| Serum phosphate, mmol/L | 55 (84.6%) | 1.1±0.4 | 1.2±0.6 (n=9) | 1.0±0.3 (n=46) | 0.716 | 0.492 |
| Average hemoglobin, g/L | 63 (96.9%) | 149.4±28.1 | 141.8±39.1 (n=11) | 151.1±25.4 (n=52) | −0.751 | 0.467 |
| White blood cells, *$10^9$/L | 62 (95.4%) | 9.2 (6.6, 13.8) | 11.8 (8.1, 14.4) (n=11) | 9.1 (6.5, 13.1) (n=51) | 0.792 | 0.428 |
| Lymphocyte percentage, % | 60 (92.3%) | 16.4 (7.6, 25.9) | 9.3±5.5 (n=11) | 20.4±12.4 (n=49) | −4.544 | <0.001 |
| Neutrophil/lymphocyte ratio | 58 (89.2%) | 5.0 (2.7, 10.9) | 10.1 (4.9, 13.3) (n=10) | 4.2 (2.5, 7.2) (n=48) | 2.285 | 0.022 |
| Serum albumin, g/L | 63 (96.9%) | 38.0 (36.4, 42.9) | 40.0 (35.9, 43.6) | 38.0 (36.7, 42.9) (n=51) | 0.403 | 0.687 |
| Blood myoglobin, ng/mL | 41 (63.1%) | 92.6 (30.7, 329.0) | 255.7 (46.8, 1200.0) (n=10) | 83.7 (30.6, 199.1) (n=31) | 1.58 | 0.119 |
| SBP, mmHg | 64 (98.5%) | 112.4±18.5 | 122.6±19.7 | 110.0±17.5 (n=52) | 2.186 | 0.033 |
| DBP, mmHg | 64 (98.5%) | 70.0 (62.5, 80.5) | 85.0 (75.0, 92.5) | 67.5 (61.0, 76.5) (n=52) | 2.531 | 0.011 |
| Heart beats per minute, bpm | 64 (98.5%) | 79.0 (68.0, 91.0) | 86.0 (82.0, 122.0) | 76.0 (66.0, 88.0) (n=52) | 2.463 | 0.014 |
| Plasma D-dimer, µg/mL | 54 (83.1%) | 0.5 (0.3, 1.1) | 0.8 (0.3, 9.1) (n=10) | 0.5 (0.3, 1.0) (n=44) | 0.613 | 0.54 |
| FAR, % | 59 (90.8%) | 6.6 (5.4, 8.6) | 7.7 (5.4, 10.9) (n=10) | 6.4 (5.4, 8.5) (n=49) | 1.172 | 0.241 |
| Direct bilirubin, mmol/L | 62 (95.4%) | 5.0 (3.7, 7.1) | 6.4 (4.9, 11.3) (n=11) | 4.9 (3.4, 7.1) (n=51) | 1.806 | 0.071 |
| Total bilirubin, mmol/L | 62 (95.4%) | 17.6 (11.2, 25.2) | 20.5 (19.2, 34.1) (n=11) | 15.9 (11.1, 23.9) (n=51) | 1.907 | 0.056 |

Note: AKI, acute kidney injury; $CO_2CP$, carbon dioxide combining power; SBP, systolic blood pressure; DBP, diastolic blood pressure; bpm, beats per minute; FAR, fibrinogen-to-albumin ratio, FAR=Fibrinogen (g/L)/ serum albumin during the illness (g/L). If the average or the lowest value during the disease was not specified, the values were considered to be those at onset.

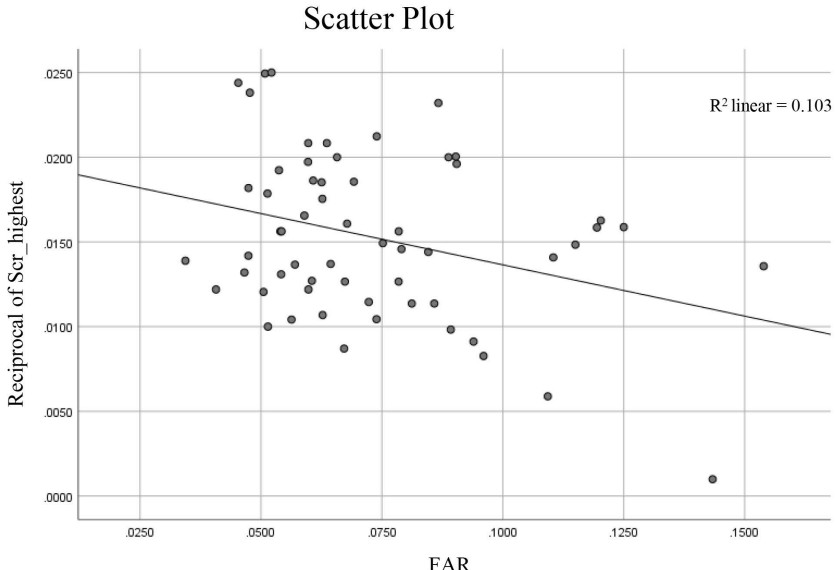

**Fig 1. The scatter plot of fibrinogen-to-albumin ratio (FAR) vs reciprocal of the highest serum creatinine concentration (Scr_highest) during the disease.**

Multivariate linear regression analysis was performed to investigate risk factors for the highest serum creatinine concentration. The reciprocal conversion caused the residual creatinine concentration to be normally distributed after regression. The independent variables included the following: Model 1: FAR. Model 2: FAR plus sex (male, 1; female, 0), age, length of stay, systolic and diastolic blood pressure, and the mean heart rate. Model 3: NLR, serum uric acid and phosphate concentrations, serum myoglobin, proteinuria positive or not, average TBIL and hemoglobin plus those in Model 2. The variance inflation factor values for all variables across the models were consistently below the commonly – accepted threshold of 10, which strongly suggests the absence of severe multicollinearity among the predictor variables. It demonstrated that FAR was associated negatively with the reciprocal of highest serum creatinine, even after adjustment by other demographic or laboratory variables (Table 2). According to the results of Model 3, the reciprocal of the highest SCr concentration ($\mu$mol/L) = 0.032–0.002 * (serum uric acid ($\mu$mol/L)/ 100) – 0.005 * serum phosphate (mmol/L) −0.078 * FAR + $\varepsilon$

## Discussion

This study showed that poisoned patients with AKI, defined according to the KDIGO criteria, in the plateau area had a lower lymphocyte percentage, higher NLR, higher blood pressure and greater heart rate in addition to significant differences in renal function indices, proteinuria and length of hospital stay. Multivariate linear regression analysis revealed that the highest SCr concentration during the course was positively related to the FAR, the concentrations of serum uric acid and phosphate. A 1%-unit increase in FAR corresponds to a 0.078% decrease in the reciprocal of the highest SCr concentration, indicating that the serum creatinine levels were elevated concomitantly. Taken together, these findings suggested that microinflammation participates in the poison-related AKI process in the plateau area.

Serum fibrinogen, a soluble glycoprotein, is a biomarker of coagulation and inflammation [13]. Thrombin has numerous downstream proteolytic targets that impact inflammatory processes. Extravascular fibrin(ogen) deposits within and around inflammatory foci suggest that fibrin may directly influence inflammatory cell locally. Indeed, in vitro analysis have shown that fibrinogen can alter leukocyte function, leading to changes in cell movement, phagocytosis, degranulation etc, which may be mediated by engagement of several leukocyte receptors and $\beta$2 integrins [14]. Albumin has anti-inflammatory

**Table 2. Multivariable linear regression analysis of risk factors for reciprocal of the highest serum creatinine concentration during the illness.**

| Model | | nonnormalized | | Normalized | t | P value | 95.0% confidence interval for B | |
|---|---|---|---|---|---|---|---|---|
| | | B | SE | Beta | | | lower limit | upper limit |
| Model 1 | (Constant) | 0.020 | 0.002 | | 10.313 | <0.001 | 0.016 | 0.023 |
| | FAR | −0.063 | 0.025 | −0.320 | −2.553 | 0.013 | −0.112 | −0.014 |
| Model 2 | (Constant) | 0.02 | 0.002 | | 12.477 | <0.001 | 0.017 | 0.024 |
| | FAR | −0.043 | 0.021 | −0.222 | −2.041 | 0.046 | −0.086 | −0.001 |
| | Sex (male,1; female:0) | −0.005 | 0.001 | −0.520 | −4.785 | <0.001 | −0.008 | −0.003 |
| Model 3 | (Constant) | 0.032 | 0.004 | | 7.973 | <0.001 | 0.024 | 0.040 |
| | FAR | −0.078 | 0.030 | −0.400 | −2.577 | 0.016 | −0.141 | −0.016 |
| | Serum uric acid * 0.01 | −0.002 | 0.001 | −0.424 | −2.812 | 0.009 | −0.003 | 0.000 |
| | Serum phosphate | −0.005 | 0.002 | −0.399 | −2.572 | 0.016 | −0.009 | −0.001 |

Note: SE, standard error; FAR, the fibrinogen-to-albumin ratio.

and antioxidant properties [9,13]. Both elevated fibrinogen concentrations and hypoalbuminemia increase blood viscosity, resulting in endothelial dysfunction or microvascular thrombosis [9], and might be associated with a poorer renal prognosis in patients with type 2 diabetes mellitus and diabetic kidney disease (DKD) [13]. The FAR has become a new type of microinflammatory index [15] and a prognostic factor for various cardiovascular diseases, DKD or AKI progression [9,13]. To date, we were unable to locate prior reports on the significance of FAR in toxic AKI.

FAR is easier to obtain than other indicators of kidney injury such as kidney injury molecular-1 (KIM-1) or neutrophil gelatinase-associated lipocalin (NGAL) etc. As biomarkers of tubular injury, KIM-1 etc are released from the proximal tubule while NGAL is released from the distal tubule [16]. Although NGAL/KIM-1 has high specificity and may reflect location of renal injury [17,18], they are not routine tests in clinical practice. So, additional detection is required and the cost is relatively high. Moreover, false positives of NGAL may occur in sepsis-related AKI [19], while the baseline level of KIM-1 is elevated in chronic kidney disease [20]. FAR integrates the dual pathways of inflammation and coagulation, which may more comprehensively reflect the systemic pathophysiological background of AKI. However, FAR may be interfered by non-renal factors (such as liver cirrhosis and malnutrition), and it needs to be interpreted in combination with the clinical background [21].

NLR, a readily available marker of inflammation and stress, has gained increasing attention as a universal marker in AKI patients [22]. The NLR was reported to be a diagnostic biomarker for AKI in severe sepsis [23,24], postsurgery [23,25,26] and radiological procedures [22,27] etc. It is also prognostic for mortality in chronic kidney disease patients [28,29]. However, there is a lack of prospective randomized trials assessing its prognostic ability [22]. Its utility in toxic AKI area is limited. A study of patients with severe acute pesticide poisoning (n = 56) revealed that the NLR was positively correlated with SCr and cystatin C etc [23]. In our study, even though we observed a significant difference between AKI and non-AKI group, the NLR did not enter the final model in the multivariable regression analysis. NLR was mainly affected by sepsis, acute phase of infection, or hematological disease, which is not manifest in this group (only 8 patients with acute infection among the 65 cases). The plateau-specific effects or small sample size factor may have some influence.

It has been reported that AKI is associated with systemic and intrarenal inflammation [1,30,31]. Pesticides can activate immune-active cells and secrete a quantity of inflammatory mediators resulting in organ dysfunction [32,33]. In patients with acute glyphosate poisoning (n = 73), leukocyte and coagulation time etc. were reported to be prognostic for mortality [7]. Oxidative stress, interstitial nephritis, direct nephrotoxicity, tubular ischemia, and rhabdomyolysis have been demonstrated to be involved in drug or diquat related kidney injury [1,34,35]. It's postulated that the oxidative stress, rhabdomyolysis, direct nephrotoxicity leading to renal epithelial cell necrosis also participates in the mushroom related kidney injury [2,5].

It is easily understood that uric acid and phosphate, excreted mainly by the kidney, were associated with the highest SCr in this group of toxic patients.

As for the type of poisons, drugs poisoning accounts for 53.8%, higher than literature (21.92%) [36]. Antipsychotics/sedatives and traditional herbs (Tibetan medicine) were main medication poisons (>50%). The public needs to be well informed to minimize the chance of exposure to excessive medicine, wild vegetables, mushrooms, and pesticides. Tibetan medicine is unique to plateau region, and its identification and processing are very crucial.

Exposure to high altitude may induce activation of hypoxia-inducible factors and eliciting reactive oxygen species. Hypoxia-inducible factors promote the expression of erythropoietin, vascular endothelial growth factor etc, promoting erythropoiesis and angiogenesis. However, we didn't find any difference in the hemoglobin levels between AKI and non-AKI group. Excessive reactive oxygen species production leads to oxidative stress and scavenging of vascular nitric oxide bioavailability, which leads to vascular endothelial dysfunction [37]. Erythropoiesis as well as platelet adhesiveness, endo-thelial dysfunction, oxygen stress, and factors promoting hemoconcentration contributed to elevated risk of thrombogene-sis in high altitude residents, especially above 3500 m [37,38]. Increased vascular permeability was postulated to reduce albumin levels due to leakage into interstitial spaces. Meanwhile, increased plasma levels of fibrinogen reported [10,11] suggests higher FAR may be induced by high-altitude hypobaric hypoxia.

Hypoxia caused macrophage accumulation in the renal tissue and renal fibroblasts activation, increasing the deposi-tion of extracellular matrix. Maladjustments to hypoxia may result in a variety of conditions including hypertension, kidney injury, etc [12]. Intermittent hypobaric hypoxia exposure was reported to induce oxidative stress along with kidney and liver injury, unusual expression of the uric acid synthesis/ excretion regulator and inflammatory response, resulting in hyperuri-cemia [39]. All these may contribute to kidney injury related to high altitude exposure.

The high proportion (60%) of intentional self-poisoning in this group requires more attention to psychological health and risk of exposure to pesticides and antipsychotic sedatives. It was reported [40] that hypobaric hypoxia at moderate-high altitude (1655 m) induces persistent endophenotypes of self-directed suicidal violence including biological signatures of inflammation, anhedonia, and depressive-like behavioral responses (inactive or increased immobility) in rats. Routine mental health and suicide risk screening in primary care and relevant community settings were recommended for patients residing at high altitude areas. For high self- harm risk population, appropriate mental health services and supervision should be strengthened in caregiver and family. Second, pesticide and antipsychotic sedative access requires tighter con-trol and monitoring. Both the sales end and the client end need to strengthen regulations on sales, distribution, packaging, safe storage (locked boxes), and to monitor unusual dispensing patterns.

## Limitations

This study has several limitations. First, this was a retrospective cohort study. Despite adjustment for potential confound-ing factors, potential selection bias and residual bias might still exist.Second, potential biases may be introduced by missing data in the laboratory results, toxin dose and comorbidity etc. Third, it is worth noticing that these findings should be interpreted with caution due to our relatively small sample size which may have risk of overfitting and other unidentified variables such as data about hypoxia or duration of high-altitude residence prior to assessment.

## Conclusion

The public needs to be well informed to minimize the chance of exposure to excessive medicine, wild vegetables, toxic mushrooms, and pesticides. Traditional Tibetan medicine, unique to this plateau region, requires professional guidance for its identification, processing, and dosage. Mental well-being in plateau areas must be prioritized, and regulatory oversight of pesticides and antipsychotic sedatives must be strengthened to mitigate the risks of overdose and toxic exposure. Microinflammation may participate in the toxic AKI. The serum creatinine concentration in poisoned patients in plateau regions may be affected by the FAR, etc., which needs further prospective studies to confirm. Therapy against microin-flammation, higher uric acid and phosphate levels may prevent further kidney injury.

## Supporting information

**S1 Table. Table 1.** Comparison of demographic and laboratory characteristics between poisoned AKI patients (n = 8) defined without regard to urine output criterion (AKI-2) and non-AKI-2 patients admitted to the hospital. Note: AKI, acute kidney injury; SBP, systolic blood pressure; DBP, diastolic blood pressure; bpm, beats per minute; AST, aspartate aminotransferase; ALT, alanine aminotransferase. If the mean/ average or the highest or lowest value during the disease was not specified, the values were considered to be the laboratory result at onset. **Table 2.** Results of Spearman bivariate correlation analysis for the highest serum creatinine concentration during the illness.
(DOCX)

**S1 File. Strobe.**
(PDF)

**S2 File. Dataset.**
(SAV)

## Author contributions

**Conceptualization:** WenLing Yang.

**Data curation:** WenLing Yang, Lixia Cui, Jianbin Zhang.

**Formal analysis:** WenLing Yang, Lixia Cui.

**Funding acquisition:** WenLing Yang.

**Investigation:** WenLing Yang, Lixia Cui, Yan He, Yong A, Quzhen Suolang, Luobu Ciren, Lei Zhang.

**Methodology:** WenLing Yang.

**Project administration:** WenLing Yang.

**Resources:** WenLing Yang, Lixia Cui, Yan He, Yong A, Jianbin Zhang, Quzhen Suolang, Luobu Ciren, Lei Zhang.

**Supervision:** WenLing Yang.

**Validation:** WenLing Yang.

**Visualization:** WenLing Yang.

**Writing – original draft:** WenLing Yang.

**Writing – review & editing:** WenLing Yang, Lixia Cui, Yan He, Yong A, Jianbin Zhang, Quzhen Suolang, Luobu Ciren, Lei Zhang.

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
