## [Decision Letter · Decision Letter 0]

9 Apr 2025

Dear Dr. Yang,

Thank you for submitting your manuscript to PLOS ONE. After careful consideration, we feel that it has merit but does not fully meet PLOS ONE’s publication criteria as it currently stands. Therefore, we invite you to submit a revised version of the manuscript that addresses the points raised during the review process.

We look forward to receiving your revised manuscript.

Kind regards,

Mohammad Barary

Academic Editor

PLOS ONE

Journal Requirements:

“This work was supported in part by a research Initiation Fund Sponsored by Peking University Third Hospital for Excellent Returned Overseas Scholars (BYSYLXHG2019007).”

4. In this instance it seems there may be acceptable restrictions in place that prevent the public sharing of your minimal data. However, in line with our goal of ensuring long-term data availability to all interested researchers, PLOS’ Data Policy states that authors cannot be the sole named individuals responsible for ensuring data access (http://journals.plos.org/plosone/s/data-availability#loc-acceptable-data-sharing-methods ).

5. Please include captions for your Supporting Information files at the end of your manuscript, and update any in-text citations to match accordingly. Please see our Supporting Information guidelines for more information: http://journals.plos.org/plosone/s/supporting-information .

Reviewers' comments:

Reviewer's Responses to Questions

**Comments to the Author**

1. Is the manuscript technically sound, and do the data support the conclusions?

Reviewer #1: Yes

Reviewer #2: Partly

2. Has the statistical analysis been performed appropriately and rigorously?

Reviewer #1: Yes

Reviewer #2: Yes

3. Have the authors made all data underlying the findings in their manuscript fully available?

Reviewer #1: Yes

Reviewer #2: No

4. Is the manuscript presented in an intelligible fashion and written in standard English?

Reviewer #1: Yes

Reviewer #2: Yes

Reviewer #1: Reviewer comments and suggestions regarding the manuscript entitled “Higher fibrinogen-to-albumin ratio predicts the severity of toxic kidney injury in patients of the plateau area—a retrospective analysis” with the manuscript number “PONE-D-24-26186.” 

This  retrospective  analysis  investigates  the  role  of  the  fibrinogen-to-albumin  ratio  (FAR)  in predicting the severity of toxic kidney injury among poisoned patients in high-altitude (plateau) regions.  Applying  KDIGO  acute  kidney  injury  criteria  and  performing  multivariate  linear regression analyses on clinical and laboratory data, the study finds that a higher FAR—along with elevated serum uric acid and phosphate levels—is significantly associated with worse renal function (as measured by serum creatinine). These findings suggest that as reflected by FAR, micro-inflammation could serve as a prognostic biomarker in this unique clinical setting.  

The  paper  can  have  an  essential contribution  to  its  field,  but  authors  should  address  some significant considerations: 

I.  Title: 

The informative title accurately reflects the study’s focus on FAR and its predictive value for 

toxic kidney injury in the plateau area. 

II.  Abstract: 

The abstract summarizes the background, methods, key results, and conclusions. It efficiently communicates the central finding that FAR and other factors correlate with increased serum creatinine levels and, hence, more severe kidney injury. 

III.  Introduction: 

The introduction sets the context by highlighting the variability in poisoning-related kidney 

injury and the limited use of AKI criteria in high-altitude regions. It justifies exploring novel 

biomarkers like FAR in this specific setting. A more detailed discussion on the physiological 

relevance of FAR in inflammation and kidney injury could further enhance this section. 

IV.  Materials and Methods: 

The methods section is comprehensive, detailing the retrospective design, inclusion criteria, data sources (electronic medical records), and statistical techniques (e.g., independent t-tests, chi-square tests, Spearman correlation, and multivariate regression). The description of the 

analytical  approach  for  determining  risk  factors  is  clear.  Additional  details  on  handling missing data would help understand the study’s robustness. 

V.  Results: 

The results are well-organized and present the demographic and clinical characteristics of the 

study population. 

VI.  Discussion: 

The discussion integrates the findings with existing literature on poisoning, inflammation, 

and kidney injury. The authors elaborate on the potential mechanisms linking FAR to renal 

dysfunction and underscore the importance of microinflammation in toxic AKI.

VII.  Limitation: 

Limitations, such as the retrospective design and small sample size, are acknowledged. A 

more in-depth exploration of potential confounders and regional factors (e.g., high-altitude physiology) would strengthen the discussion. 

VIII.  Conclusion: 

The  conclusions  succinctly  summarize  the  study’s  findings  and  their  implications.  The 

manuscript effectively suggests that FAR could be a practical, easily obtainable biomarker 

for assessing kidney injury severity in poisoned patients. The call for further prospective 

studies  to  validate  these  findings  is  appropriate  and  highlights  the  need  for  continued research. 

IX.  References: 

The comprehensive and current reference list supports the study’s context and findings. 

X.  Declaration: 

Declarations regarding funding, ethics, and competing interests are appropriately stated and align with the journal’s requirements. 

XI.  Tables: 

The tables are detailed and well-structured, providing clear comparisons between patients 

with and without AKI. They effectively highlight statistically significant differences (e.g., 

higher urea and creatinine in the AKI group) and the correlations between key biomarkers. 

Supplementary  materials  further  support  the  findings.  The  regression  model  results  are 

clearly stated, with coefficients and confidence intervals enhancing the credibility of the analysis.

Reviewer #2: Dear Authors,

Thank you for submitting your manuscript, "Higher fibrinogen-to-albumin ratio predicts the severity of toxic kidney injury in patients of the plateau area—a retrospective analysis," to PLOS ONE. Your study provides valuable insights into the prognostic role of the fibrinogen-to-albumin ratio (FAR) in toxic kidney injury within a unique high-altitude population. The retrospective design and focus on plateau physiology offer novel contributions to the literature. However, several methodological and analytical aspects require clarification to strengthen the manuscript. Below, I provide detailed recommendations to enhance the scientific rigor, clarity, and clinical relevance of your work.

Major Comments

1. Title and Abstract

• Title: Replace "toxic kidney injury" with "acute kidney injury" (AKI) to align with KDIGO criteria. Specify the altitude range (e.g., "high-altitude [3650 m] population") to better characterize your study population.

• Abstract:

o De-emphasize the neutrophil-to-lymphocyte ratio (NLR), as it was not significant in final models, and instead highlight the clinical implications of FAR (e.g., risk stratification or therapeutic decision-making).

o Correct "fulfil" to "fulfill" for consistency with American English.

o Clarify the clinical significance of the FAR coefficient (−0.078) in practical terms (e.g., how this translates to changes in creatinine levels).

2. Introduction

• Rationale: Strengthen the link between FAR and toxic AKI by citing specific mechanisms (e.g., fibrinogen’s role in microvascular thrombosis or hypoalbuminemia’s contribution to oxidative stress).

• Plateau relevance: Expand on how high-altitude physiology (e.g., chronic hypoxia) might uniquely influence FAR or AKI risk.

• Hypothesis: State explicitly (e.g., "We hypothesized that FAR would predict AKI severity independent of traditional inflammatory markers").

3. Methods

• Participants:

o Justify the exclusion of patients lacking both urine tests and renal function data, as this may introduce selection bias.

o Provide more details on the altitude range (3650 m) and its physiological implications (e.g., effects on inflammatory markers or renal hemodynamics).

• AKI definition: Clarify how baseline creatinine was determined for patients without prior measurements (e.g., back-calculation, population median).

• Statistics:

o Explain the rationale for the reciprocal transformation of creatinine, including supporting normality tests.

o Justify variable selection in multivariate models (e.g., why NLR was excluded despite univariate significance).

o Report variance inflation factors to assess multicollinearity.

4. Results

• Tables:

o Resolve discrepancies between text and tables (e.g., proteinuria cases: 17/59 in text vs. 15/59 in Table 1).

o Add interquartile ranges for median values in Table 1 and confidence intervals for beta coefficients in Table 2.

o Streamline redundant data (e.g., merge Table 1 and Supplementary Tables).

• Clinical interpretation: Translate the FAR coefficient (−0.078) into clinically meaningful terms (e.g., "A 1-unit increase in FAR corresponds to a X% increase in creatinine").

• Bias: Discuss the high rate of intentional self-poisoning (60%) and its potential impact on generalizability.

5. Discussion

• Mechanisms: Elaborate on how FAR might mediate toxic AKI (e.g., endothelial dysfunction, microvascular thrombosis). Compare FAR’s performance to established biomarkers (e.g., NGAL, KIM-1).

• NLR discrepancy: Address why NLR was non-significant despite prior literature (e.g., plateau-specific effects, small sample size).

• Limitations:

o Emphasize the small AKI subgroup (n=12) and risk of overfitting.

o Discuss unmeasured confounders (e.g., toxin dose, comorbidities).

6. Conclusions

• Actionable steps: Propose prospective validation of FAR in larger cohorts or explore therapeutic implications (e.g., targeting fibrinogen/albumin pathways).

Minor Comments

• Tables/Figures:

o Add a scatterplot of FAR vs. creatinine to visualize the relationship.

o Standardize formatting (e.g., "n (%)", spaces around "±").

• Language:

o Correct British spellings (e.g., "analyse" → "analyze").

o Define abbreviations at first use (e.g., KDIGO).

This study addresses an important clinical question with potential implications for AKI management in high-altitude populations, but requires major revision to address the methodological and analytical issues noted above. With these improvements, your manuscript could make a valuable contribution to the literature on AKI biomarkers. I would be pleased to review a revised version that addresses these concerns.

**Do you want your identity to be public for this peer review?** For information about this choice, including consent withdrawal, please see our Privacy Policy

Reviewer #1: No

Reviewer #2: **Yes: ** Bardia Karim

---

## [Author Response · Author response to Decision Letter 1]

27 May 2025

Dear reviewers:

We sincerely appreciate your insightful feedback and constructive suggestions regarding the manuscript. We have replied every question or advice step by step according to the order as the following.

Reviewers' comments:

Reviewer's Responses to Questions

Comments to the Author

1. Is the manuscript technically sound, and do the data support the conclusions?

Reviewer #1: Yes

Reviewer #2: Partly

Reply: Thanks for your support.

2. Has the statistical analysis been performed appropriately and rigorously?

Reviewer #1: Yes

Reviewer #2: Yes

Reply: Thanks for your support.

3. Have the authors made all data underlying the findings in their manuscript fully available?

Reviewer #1: Yes

Reviewer #2: No

Reply: Following consultations with the relevant hospital departments, we are able to provide the minimal data to made all data underlying the findings in their manuscript fully available.

4. Is the manuscript presented in an intelligible fashion and written in standard English?

Reviewer #1: Yes

Reviewer #2: Yes

Reply: Thanks for your support.

5. Review Comments to the Author

Reviewer #1: Reviewer comments and suggestions regarding the manuscript entitled “Higher fibrinogen-to-albumin ratio predicts the severity of toxic kidney injury in patients of the plateau area—a retrospective analysis” with the manuscript number “PONE-D-24-26186.”

This retrospective analysis investigates the role of the fibrinogen-to-albumin ratio (FAR) in predicting the severity of toxic kidney injury among poisoned patients in high-altitude (plateau) regions. Applying KDIGO acute kidney injury criteria and performing multivariate linear regression analyses on clinical and laboratory data, the study finds that a higher FAR—along with elevated serum uric acid and phosphate levels—is significantly associated with worse renal function (as measured by serum creatinine). These findings suggest that as reflected by FAR, micro-inflammation could serve as a prognostic biomarker in this unique clinical setting.

The paper can have an essential contribution to its field, but authors should address some significant considerations:

I. Title:

The informative title accurately reflects the study’s focus on FAR and its predictive value for

toxic kidney injury in the plateau area.

Reply: Thanks.

II. Abstract:

The abstract summarizes the background, methods, key results, and conclusions. It efficiently communicates the central finding that FAR and other factors correlate with increased serum creatinine levels and, hence, more severe kidney injury.

Reply: Thanks.

III. Introduction:

The introduction sets the context by highlighting the variability in poisoning-related kidney

injury and the limited use of AKI criteria in high-altitude regions. It justifies exploring novel

biomarkers like FAR in this specific setting. A more detailed discussion on the physiological

relevance of FAR in inflammation and kidney injury could further enhance this section.

Reply: The physiological relevance of FAR in inflammation and kidney injury was discussed in the second paragraph of the discussion section. As requested, I added the following sentence about the physiological relationship of FAR in inflammation and kidney injury in the introduction section:

“Emerging evidence suggests that fibrinogen-to-albumin ratio (FAR) [9], as novel inflammatory biomarkers, is significantly associated with AKI, potentially mediated through their reflection of microinflammatory states, oxidative stress pathways, and endothelial dysfunction implicated in both glomerular and tubular damage”.

IV. Materials and Methods:

The methods section is comprehensive, detailing the retrospective design, inclusion criteria, data sources (electronic medical records), and statistical techniques (e.g., independent t-tests, chi-square tests, Spearman correlation, and multivariate regression). The description of the

analytical approach for determining risk factors is clear. Additional details on handling missing data would help understand the study’s robustness.

Reply: Thanks for your advice. For variables with missing data, we report the amount how many patients had this information. Ex: For missing data in urine volume or proteinuria, we provide the description with the number how many patients had this information in table 1 as 1150.0 (750.0, 1670.0) ml/d (n= 42) or 9 (18.8%) (n= 48) in non-AKI group (n= 53), which suggest that in the 53 patients without AKI, only 42 patients had urine volume information and 48 patients had information of whether the patient had proteinuria.

To make full use of the data with missing values such as total bilirubin and hemoglobin etc, we used the averaged value of that at disease onset and that at discharge in the comparison analysis. Pairwise deletion was utilized in correlation assessments. In the multivariate linear regression analysis, we designated the highest SCr value rather than the value at onset or discharge as the dependent variable.

We added the above additional details on handling missing data in the paragraph about the statistics in Method section.

In addition, we added the shortcoming in the limitation section in lines 250: “Potential biases may be introduced by missing data in the laboratory results, toxin dose and comorbidity etc.”

V. Results:

The results are well-organized and present the demographic and clinical characteristics of the

study population.

Reply: Thanks.

VI. Discussion:

The discussion integrates the findings with existing literature on poisoning, inflammation,

and kidney injury. The authors elaborate on the potential mechanisms linking FAR to renal

dysfunction and underscore the importance of microinflammation in toxic AKI.

Reply Thanks for your understanding and support.

VII. Limitation:

Limitations, such as the retrospective design and small sample size, are acknowledged. A

more in-depth exploration of potential confounders and regional factors (e.g., high-altitude physiology) would strengthen the discussion.

Reply: Thanks for your suggestion. So, we made the following revision:

1) Even though no statistically significant difference/ relationship, we added a row of the average altitude in table 1 and S2 table.

2) We added two paragraphs in discussion section before the limitation part about the physiology or pathophysiology effect relating to high altitude.

“Exposure to high altitude may induce activation of hypoxia-inducible factors and eliciting reactive oxygen species. Hypoxia-inducible factors promote the expression of erythropoietin, vascular endothelial growth factor etc, promoting erythropoiesis and angiogenesis. However, we didn’t find any difference in the hemoglobin levels between AKI and non-AKI group. Excessive reactive oxygen species production leads to oxidative stress and scavenging of vascular nitric oxide bioavailability, which leads to vascular endothelial dysfunction [31]. Erythropoiesis as well as platelet adhesiveness, endothelial dysfunction, oxygen stress, and factors promoting hemoconcentration contributed to elevated risk of thrombogenesis in high altitude residents, especially above 3500 m [31.32]. Increased vascular permeability was postulated to reduce albumin levels due to leakage into interstitial spaces. Meanwhile, increased plasma levels of fibrinogen reported [10, 11] suggests higher FAR may be induced by high-altitude hypobaric hypoxia.

Hypoxia caused macrophage accumulation in the renal tissue and renal fibroblasts activation, increasing the deposition of extracellular matrix. Maladjustments to hypoxia may result in a variety of conditions including hypertension, kidney injury, etc [12]. Intermittent hypobaric hypoxia exposure was reported to induce oxidative stress along with kidney and liver injury, unusual expression of the uric acid synthesis / excretion regulator and inflammatory response, resulting in hyperuricemia [34]. All these may contribute to kidney injury related to high altitude exposure.”

3) We add the content in limitation section as the following: “Third, it is worth noticing that these findings should be interpreted with caution due to our relatively small sample size which may have risk of overfitting and other unidentified variables. such as data about hypoxia or duration of high-altitude residence prior to assessment.”

VIII. Conclusion:

The conclusions succinctly summarize the study’s findings and their implications. The

manuscript effectively suggests that FAR could be a practical, easily obtainable biomarker

for assessing kidney injury severity in poisoned patients. The call for further prospective

studies to validate these findings is appropriate and highlights the need for continued research.

Reply Thanks for your positive remark.

IX. References:

The comprehensive and current reference list supports the study’s context and findings.

Reply Thanks for your positive remark.

X. Declaration:

Declarations regarding funding, ethics, and competing interests are appropriately stated and align with the journal’s requirements.

Reply Thanks for your positive remark.

XI. Tables:

The tables are detailed and well-structured, providing clear comparisons between patients

with and without AKI. They effectively highlight statistically significant differences (e.g.,

higher urea and creatinine in the AKI group) and the correlations between key biomarkers.

Supplementary materials further support the findings. The regression model results are

clearly stated, with coefficients and confidence intervals enhancing the credibility of the analysis.

Reply Thanks for your positive remark.

Reviewer #2: Dear Authors,

Thank you for submitting your manuscript, "Higher fibrinogen-to-albumin ratio predicts the severity of toxic kidney injury in patients of the plateau area—a retrospective analysis," to PLOS ONE. Your study provides valuable insights into the prognostic role of the fibrinogen-to-albumin ratio (FAR) in toxic kidney injury within a unique high-altitude population. The retrospective design and focus on plateau physiology offer novel contributions to the literature. However, several methodological and analytical aspects require clarification to strengthen the manuscript. Below, I provide detailed recommendations to enhance the scientific rigor, clarity, and clinical relevance of your work.

Major Comments

1. Title and Abstract

• Title: Replace "toxic kidney injury" with "acute kidney injury" (AKI) to align with KDIGO criteria. Specify the altitude range (e.g., "high-altitude [3650 m] population") to better characterize your study population.

Reply: Thanks for your advice. I have revised the title as you suggested.

• Abstract:

o De-emphasize the neutrophil-to-lymphocyte ratio (NLR), as it was not significant in final models, and instead highlight the clinical implications of FAR (e.g., risk stratification or therapeutic decision-making).

Reply: I have corrected the abstract as the following.

“Recent evidence suggests that the fibrinogen-to-albumin ratio is significantly associated with the incidence and severity of acute kidney injury, highlighting its potential clinical utility for risk stratification and prognostication in high-risk populations. We investigated the clinical characteristics and risk factors of toxic kidney injury in highland areas using the acute kidney injury criteria defined by KDIGO, with focus on the significance of the fibrinogen-to-albumin ratio.”

o Correct "fulfil" to "fulfill" for consistency with American English.

Reply: Thank you. I have corrected as advised.

o Clarify the clinical significance of the FAR coefficient (−0.078) in practical terms (e.g., how this translates to changes in creatinine levels).

Reply: We added the following at the end of the results of the abstract: “The study demonstrated a dose-dependent association between elevated fibrinogen-to-albumin ratio and increased serum creatinine levels”.

As for the practical terms of the relationship between FAR and the highest serum creatine concentration, we prepared the following table to illustrate the specific relationship, in which we suppose the uric acid concentration was 302.9umol/L and the serum phosphate level was 1.06mmol/L, the mean values in this study.

We calculated the highest serum creatinine concentration with the formula in the manuscript, from which we can see the highest serum creatinine concentration increases with FAR. The median (interquartile) of FAR values was 6.6 (5.4, 8.6) % or 0.066 (0.054, 0.086) in this study.

For example, we suppose FAR is 6.6% (=0.066), then the reciprocal of the highest SCr concentration (µmol/L) was 0.032 - 0.002* (302.9 (µmol/L) /100)- 0.005* 1.06 (mmol/L) -0.078* 0.066 = 0.015494. Thereby, the highest SCr concentration was 1 / 0.015494 = 64.541umol/L.

FAR value (%) The highest SCR concentration, umol/l

1 50.347

2 52.405

3 54.639

4 57.071

5 59.730

5.4 60.864

6.4 63.898

6.6 64.541

7.7 68.325

8.5 71.367

10.9 143.021

2. Introduction

• Rationale: Strengthen the link between FAR and toxic AKI by citing specific mechanisms (e.g., fibrinogen’s role in microvascular thrombosis or hypoalbuminemia’s contribution to oxidative stress).

Reply The following revision was made as advised.

“Emerging evidence suggests that fibrinogen-to-albumin ratio (FAR) [9], as novel inflammatory biomarkers, is significantly associated with AKI, potentially mediated through their reflection of microinflammatory states, oxidative stress pathways, and endothelial dysfunction implicated in both glomerular and tubular damage.”

• Plateau relevance: Expand on how high-altitude physiology (e.g., chronic hypoxia) might uniquely influence FAR or AKI risk.

Reply: We added the contents in introduction as the following: “Chronic exposure to hypoxia at high altitudes was reported to increase plasma fibrinogen level [10, 11], which suggest it might elevate FAR. Hypoxia promotes fibroblast activation in renal tissue, together with inflammatory cell recruitment and tubular epithelial cell damage, leads to renal tubule interstitial fibrosis [12]”. Also, in the discussion, we added the mechanism of high-altitude physiology -FAR- AKI relationship.

• Hypothesis: State explicitly (e.g., "We hypothesized that FAR would predict AKI severity independent of traditional inflammatory markers").

Reply: Thank you for your advice. We have added this sentence in the introduction.

3. Methods

• Participants:

o Justify the exclusion of patients lacking both urine tests and renal fu

---

## [Decision Letter · Decision Letter 1]

14 Jul 2025

Dear Dr. Yang,

Thank you for submitting your manuscript to PLOS ONE. After careful consideration, we feel that it has merit but does not fully meet PLOS ONE’s publication criteria as it currently stands. Therefore, we invite you to submit a revised version of the manuscript that addresses the points raised during the review process.

We look forward to receiving your revised manuscript.

Kind regards,

Mohammad Barary, MD

Academic Editor

PLOS ONE

**Journal Requirements:**

Reviewers' comments:

Reviewer's Responses to Questions

**Comments to the Author**

Reviewer #3: All comments have been addressed

Reviewer #4: All comments have been addressed

2. Is the manuscript technically sound, and do the data support the conclusions?

Reviewer #3: Partly

Reviewer #4: Yes

3. Has the statistical analysis been performed appropriately and rigorously?

Reviewer #3: Yes

Reviewer #4: Yes

4. Have the authors made all data underlying the findings in their manuscript fully available?

Reviewer #3: Yes

Reviewer #4: Yes

5. Is the manuscript presented in an intelligible fashion and written in standard English?

Reviewer #3: Yes

Reviewer #4: Yes

**Reviewer #3:**  Study Design Clarification:

Specify the study type (cross-sectional, retrospective, etc.).

Provide detailed inclusion and exclusion criteria.

Was the sampling random, consecutive, or convenience-based?

Statistical Detail:

Include confidence intervals and effect sizes for associations.

Consider multivariate logistic regression to control for confounders (e.g., age, gender, diagnosis).

Clearly describe how the chi-square test was applied and its assumptions validated.

Presentation of Results:

Improve clarity of tables (use consistent decimal formatting, label variables clearly).

Provide total sample size in each analysis for transparency.

Strengthen the interpretation of results with comparisons to other Indian and international studies.

Expand on public health implications.

Acknowledge limitations such as potential selection bias and self-reporting of tobacco use.

**Reviewer #4: ** Thank you for making all the recommended changes, this is an interesting topic in this small population.

**Do you want your identity to be public for this peer review?** For information about this choice, including consent withdrawal, please see our Privacy Policy

Reviewer #3: No

Reviewer #4: No

---

## [Author Response · Author response to Decision Letter 2]

29 Jul 2025

Reviewers' comments:

Reviewer's Responses to Questions

Comments to the Author

1. If the authors have adequately addressed your comments raised in a previous round of review and you feel that this manuscript is now acceptable for publication, you may indicate that here to bypass the “Comments to the Author” section, enter your conflict of interest statement in the “Confidential to Editor” section, and submit your "Accept" recommendation.

Reviewer #3: All comments have been addressed

Reviewer #4: All comments have been addressed

Reply: Thank for your guidance.

2. Is the manuscript technically sound, and do the data support the conclusions?

Reviewer #3: Partly

Reviewer #4: Yes

Reply: Thank for your guidance.

3. Has the statistical analysis been performed appropriately and rigorously?

Reviewer #3: Yes

Reviewer #4: Yes

Reply: Thank for your guidance.

4. Have the authors made all data underlying the findings in their manuscript fully available?

Reviewer #3: Yes

Reviewer #4: Yes

Reply: Thank for your guidance.

5. Is the manuscript presented in an intelligible fashion and written in standard English?

Reviewer #3: Yes

Reviewer #4: Yes

Reply: Thank for your guidance.

6. Review Comments to the Author

Reviewer #3: Study Design Clarification:

Specify the study type (cross-sectional, retrospective, etc.).

Reply: Thank for your instruction. It is a retrospective study noted in the title, methods and at the second paragraph in the introduction section of the paper.

Provide detailed inclusion and exclusion criteria.

Reply: Thank for your instruction. The inclusion and exclusion criteria have been clearly demonstrated at the beginning of the methods. However, as your requirement, we modified it as the following:

“Clinical data of poisoned inpatients fulfilling the criterion were retrospectively analyzed during July 20th 2022 to July 20th 2023 to determine the clinical characteristics and risk factors of renal damage in the plateau area…

Inclusion criterion: All poisoned patients who had been hospitalized consecutively to our wards and had electronic medical records available in the past ten years were enrolled.

Exclusion criterion: The patients who did not have urine volume record and renal function data during the illness were excluded.”

Was the sampling random, consecutive, or convenience-based?

Reply: Thank for your instruction. It is consecutive, which was added into the inclusion criterion in red as above.

Statistical Detail:

Include confidence intervals and effect sizes for associations.

Reply: Thank for your instruction. The confidence intervals and effect sizes have been listed in table 2 for associations. In multivariable linear regression analysis, the absolute value of normalized coefficient---beta value was the effect size which can compare the devotion of different variables to the independent variable (the reciprocal of the highest serum creatinine abbreviated as SCr in this study). Positive value indicates the value of independent variable (the reciprocal of the highest scr in this study) will increase with the increase of this variable, while negative indicates protective or decreasing the value of independent variable (the reciprocal of the highest scr). Ex. In model 3, the effect sizes of FAR, uric acid and phosphate are close, and serum uric acid devotes a little more to the reciprocal of SCr than the other two variables (-0.424 to -0.400, -0.399 respectively).

Consider multivariate logistic regression to control for confounders (e.g., age, gender, diagnosis).

Reply: Thank for your instruction. There were only 12 cases diagnosed as AKI. Due to the limited number of events, we can only analyze the influence of 1 to 3 confounders or variables (12 divided by 5 to 10) on the dependent variable. Moreover, some confounders have missing values, then the analyzed cases in the multivariate logistic regression will be less. Taking model 3 as an example, the different variables may have missing values in different patient, then only 23 cases were analyzed in logistic regression. So, it needs more cases in the study to utilize multivariate logistic regression. However, the highest serum creatinine is a continuous variable and all 65 patients had the value. In multivariate linear regression, it allowed pairwise analysis for confounders with missing value. Since logistic regression uses a categorical dependent variable (AKI or not) which will lose some information than the quantitative variable (the reciprocal of the highest serum creatinine) in linear regression in this study. So, it is not advisable. In the multivariate linear regression, FAR has been adjusted by sex, age etc.

We added a sentence in the paragraph before Table 2 as the following. “It demonstrated that FAR was associated negatively with the reciprocal of highest serum creatinine, even after adjustment by other demographic or laboratory variables (Table 2).”

Clearly describe how the chi-square test was applied and its assumptions validated.

Reply: Thank for your instruction. We have revised as the following. “Categorical variables are compared with chi-square tests, corrected for continuity if it is a 2* 2 table or Fisher’s exact test if >20% of cells have the expected count < 5 in a table larger than 2×2.”

Presentation of Results:

Improve clarity of tables (use consistent decimal formatting, label variables clearly).

Reply: Thank for your instruction. We have revised the tables as request.

Provide total sample size in each analysis for transparency.

Reply: Thank for your instruction. We have labeled the sample size in yellow in tables.

For multivariate linear regression, I listed the relative part below (/missing pairwise, different from multivariate logistic regression which allowed only listwise for missing values):

Model 1:

n

reciprocal of scr_highest 65

FAR 59

Model 2:

n

reciprocal of scr_highest 65

sex: male,1; female: 65

Age, years 65

Length of stay, days 65

SBP at onset, mmHg 64

DBP at onset, mmHg 64

HR_average 65

FAR 59

Model 3:

n

reciprocal of scr_highest 65

sex: male,1; female:0 65

Age, years 65

Length of stay, days 65

SBP at onset, mmHg 64

DBP at onset, mmHg 64

HR_average 65

FAR 59

Serum phosphate, mmol/L 55

Blood myoglobin, ng/mL 41

NLR 58

Uric acid*0.01 44

Proteinuria or not 65

Average hemoglobin, g/L 63

Total bilirubin, mmol/L 62

Strengthen the interpretation of results with comparisons to other Indian and international studies.

Reply: We appreciate this constructive suggestion. To address this, the Discussion section has been expanded to include comparative analyses with multiple international studies, including:

• Indian research (Ref. 19)

• Multicenter collaborations (Ref. 12: Italy/India)

• Studies from Japan (Ref. 31), Australia (Ref. 29), France (Ref. 6), the United States (Refs. 2, 14, 36), and Brazil (Ref. 1).

Additionally, we conducted a systematic PubMed search for recent Indian literature relevant to our study domain. After critical evaluation of the following publications, we determined that they do not contribute substantively to the interpretation of our results:

1. Parthasarathy et al. (2021, Nephrology)

Parthasarathy R., Mathew M., Koshy P., et al. Traditional medicines prescribed for prevention of COVID-19: Use with caution. Nephrology (Carlton) 2021, 26(12): 961-964

Rationale for exclusion: Reports an isolated case of nephrotoxicity linked to Indian traditional COVID-19 prophylactics (kabasura/nilavembu kudineer), which does not align with our mechanistic focus.

2. Ralph et al. (2024, Toxicon)

Ralph R., Sharma D., Jain R., e al. Protobothrops jerdonii (Jerdon's pit viper) and Protobothrops himalayanus (Himalayan lance-headed pit viper) bites: Clinical report on envenomings from North-East India, managed through remote consultation by a national-level Poison control center. Toxicon 2024, 242:107704

Rationale for exclusion: Documents acute kidney injury (AKI) from viper envenomation—a contextually distinct etiology from our research parameters.

3. Pannu (2019, Current Drug Metabolism)

Pannu K. Methotrexate overdose in clinical practice. Curr Drug Metab 2019,20(9): 714-719

Rationale for exclusion: Focuses on methotrexate overdose toxicology, beyond our study scope.

4. Sirur et al. (2022, Wilderness & Environmental Medicine)

Sirur F.M., Balakrishnan J.M. and Lath V.. Hump-Nosed Pit Viper Envenomation in Western Coastal India: A Case Series. Wilderness Environ Med 2022, 33(4): 399-405

Rationale for exclusion: Examines hump-nosed viper envenomation syndromes in Western India, unrelated to our pathophysiological framework.

5. Renu et al. (2022, Molecules)

Renu K, Mukherjee AG, Wanjari UR, et al. Misuse of Cardiac Lipid upon Exposure to Toxic Trace Elements-A Focused Review. Molecules, 2022;27(17):5657. doi: 10.3390/molecules27175657.

Rationale for exclusion: Explores trace element-induced cardiotoxicity through lipid dysregulation, diverging from our core investigative pathways.

We confirm that all citations incorporated in the revised manuscript directly substantiate our comparative analysis. Thank you for your guidance throughout this revision process.

Expand on public health implications.

Reply: Thank for your instruction. I highlighted the content in page 19: “The high proportion (60%) of intentional self-poisoning in this group requires more attention to psychological health and risk of exposure to pesticides and antipsychotic sedatives.” And the first sentence in the conclusion was about the public implications “The public needs to be well informed to minimize the exposure to medicine, pesticide, mushroom etc”.

As requested, we expanded the contents as the following.

“Routine mental health and suicide risk screening in primary care and relevant community settings were recommended for patients residing at high altitude areas. For high self- harm risk population, appropriate mental health services and supervision should be strengthened in caregiver and family. Second, pesticide and antipsychotic sedative access requires tighter control and monitoring. Both the sales end and the client end need to strengthen regulations on sales, distribution, packaging, safe storage (locked boxes), and to monitor unusual dispensing patterns.”

Acknowledge limitations such as potential selection bias and self-reporting of tobacco use.

Reply: Thank for your instruction. As request, we acknowledge the limitations at the end of discussion section: “Despite adjustment for potential confounding factors, potential selection bias and residual bias might still exist.” Sorry, tobacco use is outside our focus in this study.

Reviewer #4: Thank you for making all the recommended changes, this is an interesting topic in this small population.

Reply: Thank for your instruction.

7. PLOS authors have the option to publish the peer review history of their article (what does this mean?). If published, this will include your full peer review and any attached files.

Do you want your identity to be public for this peer review? For information about this choice, including consent withdrawal, please see our Privacy Policy.

Reviewer #3: No

Reviewer #4: No

Reply: Thank for your instruction.

---

## [Decision Letter · Decision Letter 2]

13 Nov 2025

Dear Dr. Yang,

Thank you for submitting your manuscript to PLOS ONE. After careful consideration, we feel that it has merit but does not fully meet PLOS ONE’s publication criteria as it currently stands. Therefore, we invite you to submit a revised version of the manuscript that addresses the points raised during the review process.

We look forward to receiving your revised manuscript.

Kind regards,

Shivkumar Gopalakrishnan, MD

Academic Editor

PLOS ONE

Journal Requirements:

Reviewers' comments:

Reviewer's Responses to Questions

**Comments to the Author**

Reviewer #4: (No Response)

Reviewer #5: All comments have been addressed

2. Is the manuscript technically sound, and do the data support the conclusions?

Reviewer #4: Partly

Reviewer #5: Yes

3. Has the statistical analysis been performed appropriately and rigorously?

Reviewer #4: Yes

Reviewer #5: Yes

4. Have the authors made all data underlying the findings in their manuscript fully available?

Reviewer #4: Yes

Reviewer #5: Yes

5. Is the manuscript presented in an intelligible fashion and written in standard English?

Reviewer #4: Yes

Reviewer #5: Yes

Reviewer #4: Thank you for submitting this amended article.

I would suggest the following changes

Introduction

1. Change "food poisoning" to "ingestion of known toxic substances" (mushrooms/ fish gall cysts) may this be deliberate or accidental. Food poisoning in many countries is where a person becomes mildly/ moderately unwell often due to poor hygiene during food preparation that leads to gastrointestinal illness.

2. On line 50 you say that AKI definition varies widely- but KDGIO international guidance has been in place since 2012. Please amend to say that prior to 2012 definitions varied, and then go on to talk about the impact that had on previous research.

Methods.

3. Within the methods section you need to state how you identified these patients.

4. Why did you exclude patients who did not have their urine output recorded during their illness? How did this impact on your sample size.

5. How did you ensure that AKI based on urine output was true? It is well known that urine output documentation is poor outside of critical care areas. This needs to make very clear as a limitation as 1/3 of your AKI patients may not have had an AKI at all but only poor fluid balance documentation. You also need to state how many of these patients were determined to be AKI based on urine output alone here ( I know this is in your results section).

Results

This section is quite muddled with the results being focused on the poisoning elements (which is important but needs to be presented more clearly).

6. The first part of your results should be that only 12 patients of the 65 identified were found to have AKI. You spend a long time discussing the causes of poisoning in all patients- but as this article is about AKI in this patient group your results section should begin with the most important finding.

7. Within your results it would provide more clarity if you provided how many of the 35 patients who were "poisoned" with medication were intentional overdoses.

8. Is Hemorperfusion the same as hemofiltration? Which I suspect it is- if patients are getting this as a first line treatment it may explain why patients are developing AKI (removal of toxin = no toxins to cause AKI). This should be included in your discussion.

Discussion

9. On line 277 you use the word etc- please clarify what else is reported to be prognostic for mortality.

Conclusion

10. You have not appropriately addressed the previous reviewers comments here. More detail is required in understanding what the public needs to do. You do not want people to minize their exposure to antidepressant/ anti psychotic medications- they need to be aware of risks of toxicity of overdosing.

11. Line 336 has a statement in with no clarification or depth to justify it.

11. On line 336 you use the word etc- please clarify what else is required.

You need a clear limitations section that is separate from the discussion.

Reviewer #5: Dear Authors

please provide this additional information and modifications:

Please replace all instances of “predicts” with “is associated with” throughout the manuscript, including the title and abstract, unless you add a validated prediction framework (e.g., model development with internal validation, calibration, and performance metrics).

Please provide the assay methods and units for fibrinogen and albumin, and state the exact formula used to compute the fibrinogen-to-albumin ratio (FAR). Clarify the measurement units employed (e.g., g/L vs. mg/dL) to ensure reproducibility and interpretability.

Please define precisely when each laboratory value—including fibrinogen, albumin, and creatinine—was obtained relative to poisoning onset, emergency department presentation, and any treatments (e.g., hemoperfusion). Indicate whether values were collected before or after interventions and specify the time window used for analyses.

**Do you want your identity to be public for this peer review?** For information about this choice, including consent withdrawal, please see our Privacy Policy

Reviewer #4: No

Reviewer #5: No

---

## [Author Response · Author response to Decision Letter 3]

21 Nov 2025

PONE-D-24-26186R2 Higher fibrinogen-to-albumin ratio predicts the severity of toxin associated acute kidney injury in high-altitude (3650 m) population—a retrospective analysis PLOS ONE

Dear Dr. Yang,

Thank you for submitting your manuscript to PLOS ONE. After careful consideration, we feel that it has merit but does not fully meet PLOS ONE’s publication criteria as it currently stands. Therefore, we invite you to submit a revised version of the manuscript that addresses the points raised during the review process.

A rebuttal letter that responds to each point raised by the academic editor and reviewer(s). You should upload this letter as a separate file labeled 'Response to Reviewers'.

A marked-up copy of your manuscript that highlights changes made to the original version. You should upload this as a separate file labeled 'Revised Manuscript with Track Changes'.

An unmarked version of your revised paper without tracked changes. You should upload this as a separate file labeled 'Manuscript'.

REPLY: Thanks. We have completed the revision and uploaded.

REPLY: Thanks. No change.

REPLY: Thanks. It is not applicable.

We look forward to receiving your revised manuscript.

Kind regards,

Shivkumar Gopalakrishnan, MD

Academic Editor PLOS ONE

Journal Requirements:

REPLY: Thanks. All cited papers have been checked.

Reviewers' comments:

Reviewer's Responses to Questions

Comments to the Author

1. If the authors have adequately addressed your comments raised in a previous round of review and you feel that this manuscript is now acceptable for publication, you may indicate that here to bypass the “Comments to the Author” section, enter your conflict of interest statement in the “Confidential to Editor” section, and submit your "Accept" recommendation.

Reviewer #4: (No Response)

Reviewer #5: All comments have been addressed

REPLY: Thanks.

2. Is the manuscript technically sound, and do the data support the conclusions?

Reviewer #4: Partly

Reviewer #5: Yes

REPLY: Thanks.

3. Has the statistical analysis been performed appropriately and rigorously?

Reviewer #4: Yes

Reviewer #5: Yes

REPLY: Thanks.

4. Have the authors made all data underlying the findings in their manuscript fully available?

Reviewer #4: Yes

Reviewer #5: Yes

REPLY: Thanks.

5. Is the manuscript presented in an intelligible fashion and written in standard English?

Reviewer #4: Yes

Reviewer #5: Yes

REPLY: Thanks.

6. Review Comments to the Author

Reviewer #4: Thank you for submitting this amended article.

I would suggest the following changes

Introduction

1. Change "food poisoning" to "ingestion of known toxic substances" (mushrooms/ fish gall cysts) may this be deliberate or accidental. Food poisoning in many countries is where a person becomes mildly/ moderately unwell often due to poor hygiene during food preparation that leads to gastrointestinal illness.

REPLY: Thanks. Except one patient, all the other 11 patients ate poisonous mushroom, wild vegetables or contaminated fish by mistake and they thought them edible. To distinguish with the poisoning from medicine, pesticides, we used the “food poisoning” in general, while “ingestion of known toxic substances” may include ingestion of medicine, pesticides, household chemicals, poisonous mushroom, toxic seafood, contaminated fish etc. So, we think changing to “toxic substances eaten as food” may be more suitable to this context.

2. On line 50 you say that AKI definition varies widely- but KDGIO international guidance has been in place since 2012. Please amend to say that prior to 2012 definitions varied, and then go on to talk about the impact that had on previous research.

REPLY: Thanks. We revised it as the following: “However, the definition of AKI varies widely across the literature prior to 2012”

Methods. 3. Within the methods section you need to state how you identified these patients.

REPLY: Thanks. We have stated it clearly in the method section:

Inclusion criterion: All poisoned patients who had been hospitalized consecutively to our wards and had electronic medical records available in the past ten years were enrolled.

Exclusion criterion: The patients who had neither urine volume record nor renal function data during the illness were excluded.

4. Why did you exclude patients who did not have their urine output recorded during their illness? How did this impact on your sample size.

REPLY: Thanks. In the method section, we have stated that the exclusion criterion was those patients who did not have urine volume record and renal function data during the illness. In the result section, we have also stated clearly that: “one patient was excluded due to lack of both renal function and urine volume record”.

If the patient has neither serum creatinine data nor urine volume data, how to judge whether they have AKI according to AKI definition? To make it more clearer, we revised the sentence as “The patients who had neither urine volume record nor renal function data during the illness were excluded.”

5. How did you ensure that AKI based on urine output was true? It is well known that urine output documentation is poor outside of critical care areas. This needs to make very clear as a limitation as 1/3 of your AKI patients may not have had an AKI at all but only poor fluid balance documentation. You also need to state how many of these patients were determined to be AKI based on urine output alone here (I know this is in your results section).

REPLY: Thanks. What you said about the limitation of urine criterion is right, we have stated how to use urine volume criterion in the method: “AKI status was defined as literature [8]. An increase in serum creatinine (SCr) concentration from baseline can be substituted as a decrease from the baseline during follow-up if there are no prior tests. A urine output less than 400 mL/day may be an alternative to ≤ 0.5 mL/ (kg/h) for 6 hours. For those with only one test of serum creatinine, we made diagnosis of AKI with reference to the criteria of urine volume and clinical diagnosis.”

We also worry about the confusion of urine volume criterion for AKI, so we analyzed those defined using serum creatinine criterion only. “If we defined AKI without regard to urine output, then only 8 patients had AKI (Table 1 in S1 Tables).” Four patients who were defined as AKI according to urine volume criterion had urine volume of 170-300ml/d at admission and 1900-4850ml/d at discharge after therapy. They had suffered from confusion (n = 3) or drowsiness (n=1). All the four patients maybe had only poor fluid balance after poisoning, but if no therapy was commenced, more serious kidney injury may happen.

Results This section is quite muddled with the results being focused on the poisoning elements (which is important but needs to be presented more clearly).

REPLY: Thanks. All the 65 patients were admitted to wards because of poisoning due to different poisons at plateau, so poisoning elements may be different in some degree from those in plain areas. If there is any content not stated clearly, please don’t hesitate to tell us and we will made it clearer.

6. The first part of your results should be that only 12 patients of the 65 identified were found to have AKI. You spend a long time discussing the causes of poisoning in all patients- but as this article is about AKI in this patient group your results section should begin with the most important finding.

REPLY: Thanks. Our paper focused on two points, namely toxic kidney injury using the criterion of AKI and plateau area. All analysis about poisoning elements was to see if there was any difference in plateau compared with those in plain areas.

7. Within your results it would provide more clarity if you provided how many of the 35 patients who were "poisoned" with medication were intentional overdoses.

REPLY: Thanks. Line 146 in manuscript with marks: intentional self-poisoning: n =39; intentional medical overdoses: n=23.

We revised the results as the following:

“Further analysis of the 39 patients with intentional self-poisoning demonstrated the main types of poisons were pesticides (38.5%, n = 15) and medication (59.0%, n=23), while antipsychotic sedatives (60.9%, n = 14) were the main type of medicine.”

8. Is Hemorperfusion the same as hemofiltration? Which I suspect it is- if patients are getting this as a first line treatment it may explain why patients are developing AKI (removal of toxin = no toxins to cause AKI). This should be included in your discussion.

REPLY: Thanks. Hemoperfusion is different from hemofiltration and the former was more useful in the removal of toxins. Whether removal of toxin means no or smaller quantity of toxins to cause AKI depend on the following points: 1) How long the patient was transferred to emergency after poisoning; 2) How long the blood purification was commenced after admission; 3) The protein binding capacity of the toxins. Multiple hemoperfusion therapy may be necessary for those taking toxins with high protein binding capacity.

Among the 65 patients diagnosed as poisoning, thirty-four patients (52.3%) underwent blood purification. Among them, hemoperfusion was carried out in 33 patients for 1-4 times.

Discussion 9. On line 277 you use the word etc- please clarify what else is reported to be prognostic for mortality.

REPLY: Thanks. In patients with acute glyphosate poisoning (n=73), leukocyte and coagulation time etc. were reported to be prognostic for mortality [7]. The other factors included gamma-glutamyl transpeptidase (GGT), age and medical contact time (categories: 1, within 2h; 2, 2-6h; 3, above 6h).

Conclusion 10. You have not appropriately addressed the previous reviewers comments here. More detail is required in understanding what the public needs to do. You do not want people to minize their exposure to antidepressant/ anti psychotic medications- they need to be aware of risks of toxicity of overdosing.

REPLY: Thanks. We revised the first sentence in the conclusion section as “The public needs to be well informed to minimize the chance of exposure to excessive medicine, wild vegetables, toxic mushrooms, and pesticides. Traditional Tibetan medicine, unique to this plateau region, requires professional guidance for its identification, processing, and dosage. Mental well-being in plateau areas must be prioritized, and regulatory oversight of pesticides and antipsychotic sedatives need to be strengthened to mitigate the risks of overdose and toxic exposure.”

11. Line 336 has a statement in with no clarification or depth to justify it.

REPLY: Thanks. We revised it as: “The public needs to be well informed to minimize the chance of exposure to excessive medicine, wild vegetables, toxic mushrooms, and pesticides. Traditional Tibetan medicine, unique to this plateau region, requires professional guidance for its identification, processing, and dosage. Mental well-being in plateau areas must be prioritized, and regulatory oversight of pesticides and antipsychotic sedatives need to be strengthened to mitigate the risks of overdose and toxic exposure.”

12. On line 336 you use the word etc- please clarify what else is required.

REPLY: Thanks. Etc: wild vegetables, toxic seafood, contaminated fish or toxic substances eaten as food. We revised it as the above reply.

You need a clear limitations section that is separate from the discussion.

REPLY: Thanks. It has been separated.

Reviewer #5: Dear Authors please provide this additional information and modifications:

Please replace all instances of “predicts” with “is associated with” throughout the manuscript, including the title and abstract, unless you add a validated prediction framework (e.g., model development with internal validation, calibration, and performance metrics).

REPLY: Thanks. All have been corrected.

Please provide the assay methods and units for fibrinogen and albumin, and state the exact formula used to compute the fibrinogen-to-albumin ratio (FAR). Clarify the measurement units employed (e.g., g/L vs. mg/dL) to ensure reproducibility and interpretability.

REPLY: Thanks. Fibrinogen is assayed by the thrombin clotting method, while albumin is measured by the bromcresol green method, both are reported in g/L.

Fibrinogen-to-albumin ratio (FAR)= Fibrinogen (g/L)/ serum albumin during the illness (g/L)

This formula was added to the note for Table 1.

Please define precisely when each laboratory value—including fibrinogen, albumin, and creatinine—was obtained relative to poisoning onset, emergency department presentati

---

## [Decision Letter · Decision Letter 3]

29 Dec 2025

Higher fibrinogen-to-albumin ratio is associated with the severity of toxic kidney injury in patients of the plateau area- a retrospective analysis

PONE-D-24-26186R3

Dear Dr. WenLing Yang,

We’re pleased to inform you that your manuscript has been judged scientifically suitable for publication and will be formally accepted for publication once it meets all outstanding technical requirements.

Kind regards,

Dr Shivkumar Gopalakrishnan, MD

Academic Editor

PLOS One

Additional Editor Comments (optional): NIL

Reviewers' comments:

Reviewer's Responses to Questions

**Comments to the Author**

Reviewer #4: All comments have been addressed

Reviewer #5: All comments have been addressed

2. Is the manuscript technically sound, and do the data support the conclusions?

Reviewer #4: Yes

Reviewer #5: Yes

3. Has the statistical analysis been performed appropriately and rigorously?

Reviewer #4: I Don't Know

Reviewer #5: Yes

4. Have the authors made all data underlying the findings in their manuscript fully available?

Reviewer #4: Yes

Reviewer #5: Yes

5. Is the manuscript presented in an intelligible fashion and written in standard English?

Reviewer #4: Yes

Reviewer #5: Yes

Reviewer #4: Thank you for addressing the comments that were previously submitted. This is an interesting article

Reviewer #5: (No Response)

**Do you want your identity to be public for this peer review?** For information about this choice, including consent withdrawal, please see our Privacy Policy

Reviewer #4: No

Reviewer #5: No

---

## [Editor Report · Acceptance letter]

PONE-D-24-26186R3

PLOS One

Dear Dr. Yang,

I'm pleased to inform you that your manuscript has been deemed suitable for publication in PLOS One. Congratulations! Your manuscript is now being handed over to our production team.

Kind regards,

on behalf of

Dr. Shivkumar Gopalakrishnan

Academic Editor

PLOS One